# Graphene Oxide Modified Polyamide 66 Ultrafiltration Membranes with Enhanced Anti-Fouling Performance

**DOI:** 10.3390/membranes12050458

**Published:** 2022-04-24

**Authors:** Jiangyi Yan, Lihong Nie, Guiliang Li, Yuanlu Zhu, Ming Gao, Ruili Wu, Beifu Wang

**Affiliations:** 1College of Petrochemical Engineering and Environment, Zhejiang Ocean University, Zhoushan 316000, China; jiang1842022@163.com (J.Y.); nielihong1975@163.com (L.N.); ligl@zjou.edu.cn (G.L.); xiaoluer113@163.com (Y.Z.); gaoming97@163.com (M.G.); 2Sichuan Bureau of National Food and Strategic Reserves Administration, Chongqing 401326, China; wuruili2022@163.com

**Keywords:** polyamide ultrafiltration membrane, graphene oxide, anti-fouling, strengthening and toughening

## Abstract

Improving the contamination resistance of membranes is one of the most effective ways to address the short service life of membranes. While preparing the membrane system structure, doping nanoparticles into the polymer matrix is beneficial to the preparation of high-performance membranes. To develop a new structure for membrane contamination protection, in this study, a novel asymmetric polyamide 66 composite ultrafiltration (UF) membrane was fabricated by incorporating different masses (ranging from zero to 0.5 wt.%) of graphene oxide (GO) into the polyamide 66 microporous substrate, using formic acid and propylene carbonate as solvents. The effects of GO doping on the morphology, microporous structure and surface of ultrafiltration membranes were investigated by atomic force microscopy (AFM), scanning electron microscopy (SEM), integrated thermal analysis (DSC) and contact angle (CA). In addition, pure water flux, bovine serum albumin (BSA) rejection and contamination resistance were measured to evaluate the filtration performance of different membranes. The overall performance of all the modified membranes was improved compared to pure membranes. The results of contact angle and permeation experiments showed that the addition of GO improved the hydrophilicity of the membrane, but reduced the permeability of the membrane. The minimum flux was only 3.5 L/m^2^·h, but the rejection rate was 92.5%. Most noteworthy was the fact that GO further enhanced the anti-pollution performance of the membranes and achieved a remarkable performance of 91.32% when the GO content was 0.5 wt.%, which was 1.36 times higher than that of the pure membrane. Therefore, optimal performance was achieved. Furthermore, the UF membrane made of composite substrate offers a promising solution for the development of long-life ultrafiltration membranes with better stability, high-cost efficiency and adequate chemical durability.

## 1. Introduction

Membrane separation technology, as a new efficient separation technology, has proven to have excellent separation, concentration, purification, environmental protection, energy saving and other advantages, and has become a hot topic both at home and abroad. As one of these forms of technology, ultrafiltration membrane technology can effectively treat pollutants in water, and then realize the purification and concentration of water resources [1]. Due to its good purification effect, it can avoid excessive accumulation of impurities causing secondary pollution, play an important role in sewage treatment and other fields, and has a wide range of application prospects [2]. Since the invention of membrane technology, membrane contamination has been a challenge [3] that has adversely affected almost all membranes, causing considerable loss in purification operating costs and separation efficiency. Therefore, many successful methods have been applied in the past decades of research. For example, the choice of hydrophilic materials for membrane formation [3], the choice of hydrophilic additives to improve the physicochemical properties of the membrane, the introduction of oxygen-containing functional groups and the doping of non-toxic organic fluorescent materials with aggregation-induced emission (AIE) [4].

Through hydrophilic modification, a dense hydration layer can be formed on the surface of the film to improve anti-fouling resistance. Common hydrophilic nanomaterials include silicon dioxide (SiO_2_), titanium dioxide (TiO_2_), acidified multi-walled carbon nanotubes (A-MWCNT), graphene oxide (GO) and so on. Karimi, Atefeh, et al. used ball-milled Cu_2_S nanoparticles as a modifier for polyvinylidene fluoride (PVDF) ultrafiltration membranes, which enabled the composite membranes to highly reject anionic dyes and increased the hydrophilicity of the membranes [5]. Han, Jun-Cheng, et al. added multi-walled carbon nanotube modified particles to regenerate cellulose ultrafiltration membranes to enhance anti-fouling ability. When the cellulose membrane is coated with MWNTS, its anti-fouling ability is more than doubled and it has the potential to remove hydrophilic pollutants from water [6]. In order to improve the separation and anti-fouling performance of the membrane, Moghimifar V et al. modified the surface of the polyethersulfone (PES) ultrafiltration membrane with corona air plasma and coated the surface with TiO_2_ nanoparticles [7]. Kassa, Shewaye Temesgen, et al. applied metallic glass (MG) to the surface of polysulfone composite membrane to achieve a high pure water flux and protein retention rate [8]. Cheng, Kai, et al. prepared a novel anti-fouling polyvinylidene fluoride (PVDF) ultrafiltration membrane with enhanced permeability selectivity by co-deposition of dopamine and small molecular amphoteric ions (DMAPAPS), which showed excellent anti-bacterial activity and excellent chemical properties [9]. Polymer membranes are vulnerable to severe membrane pollution, due to the influence of their inherent material properties. By loading hydrophilic nanomaterials and distributing them in the form of a concentration gradient throughout the membrane structure, it strongly weakens interaction between pollutants and membranes, reduces costs and makes the membrane more environmentally friendly. The modified functional membrane not only shows excellent anti-pollution ability, but also combines two anti-pollution mechanisms, which is different from a single anti-pollution mechanism.

Polyamide 66 can be an excellent ultrafiltration membrane material, due to its high tensile strength and impact resistance to ensure high membrane stability [10]. These favorable properties make PA66 suitable for the preparation of wettable porous membranes and are prospective for applications in separation processes. However, PA66 ultrafiltration membranes are susceptible to contamination by contaminants during water treatment, due to their weak hydrophilic properties, which affects the sustainability of membrane purification projects.

Graphene oxide, a unique hydrophilic material, has been favored by scientists since its introduction. Due to its huge specific surface area, large layer spacing and large number of oxygen-containing functional groups on its surface, unlike the conventional type of flexible substances, and having its own antibacterial properties [11,12], combining it with polymeric materials is expected to prepare ultrafiltration membranes with excellent anti-fouling properties, which have potential applications in wastewater purification. In fact, a number of studies have shown that GO has a significant positive effect in improving membrane resistance to contamination [13].

Although there are some recent reports on GO/polymer UF membranes, we investigated, for the first time, the effect of GO on the microporous structure of polyamide 66 ultrafiltration membranes, with emphasis on the anti-fouling properties of the membranes.

The purpose of this study is to prepare a new type of ultrafiltration membrane with strong anti-fouling performance. In this paper, a polyamide 66 ultrafiltration membrane was modified with GO to improve its mechanism and microstructure. The chemical composition, microstructure and separation and interception performance of the composite ultrafiltration membrane were studied and analyzed by various analysis methods.

## 2. Experimental

### 2.1. Materials

Polyamide 66 (PA66, A.R) was provided by French manufacturer Rhodia Co., Ltd., Bangkok, Thailand; Formic acid (HCOOH, FA, 94%) was purchased from Yangzi Petrochemical-Basf Co., Ltd., Nanjing, China; 1,2-propanediol carbonate (C_4_H_6_O_3_, PC, A.R) was purchased from National Pharmaceutical Chemicals Co., Ltd., Beijing, China; Bovine serum albumin (BSA, Mn = 66,000) was purchased from West Asia Chemical Technology Co., Ltd., Jinan, China; Time nano industrial graphene oxide powder (GO, > 99 wt.%) was purchased from Chengdu Organic Chemistry Co., Ltd., Chinese Academy of Sciences, Chengdu, China.

### 2.2. Preparation of PA66-GO Flat Sheet Membranes

Four different composite membranes were prepared by the immersion phase inversion method. The compositions and codes for different samples are shown in Table 1. The preparation details of the casting solution were as follows: (1)Different amounts of GO (from 0 wt.% to 0.5 wt.%) were added into appropriate amounts of FA and PC mixed solvent. (The ratio of the two solvents was 5/1.) and the solution was treated with ultrasonic for 1 h to disperse GO and minimize agglomeration;(2)The PA66 solution was prepared in advance. PA66 (22 wt.%) was dissolved in a mixture of the remaining FA and PC. The mixture was blended at 60 °C until a clear homogeneous solution was obtained. The solution was aged for a period of 1 or 2 h in a thermostat held at 45 °C to remove air bubbles. Then the solution was naturally cooled to room temperature;(3)The GO solution was then added slowly into the PA66 mixture solution under vigorous stirring. The mixed solution of the two was stirred for 5 h to obtain a homogenous solution for casting. The prepared casting solution was sealed and stored for more than 12 h to remove air bubbles. To remove residual air bubbles trapped within the dope solutions, each solution was then subjected to 1 h of ultrasonication, followed by casting on a glass plate, using a self-made casting glass rod. Then, the cast glass plate was left for 3 min at an ambient temperature before being immersed into a water coagulation bath at room temperature for the phase inversion process to take place. Once the membrane was peeled off from the glass plate, it was transferred to another water bath and kept for 24 h to remove residual solvent. For pure PA66 membrane, the same method was applied.

Before usage, the membranes were washed with deionized water three times. The cleaned membrane was dried for subsequent characterization. The preparation process and formation mechanism of the membrane are shown in Figure 1.

### 2.3. Characterization

#### 2.3.1. GO Characterization

To evaluate the extent of dispersion of the GO in different solvents, a UV/visible spectroscopy (TU-1901, General analysis) was used. The X-ray diffraction (XRD) measurement of GO was carried out with an X-ray diffractometer with Cu Kα radiation (λ = 0.154 nm, D/max-rB 12 kW Rigaku) which operated at 30 mA and 40 kV from 5° and 80° with a step increment rate of 1.20°/min. Melting behavior was determined using a 200-F3 differential scanning calorimeter (DSC, NETZSCH, Germany). About 8 mg GO solid powder was sealed up in an aluminum pan. It was kept at 20 °C for 3 min, then heated to 400 °C at a heating rate of 10 °C/min and maintained 3 min to remove the thermal history, and then chilled to 20 °C (the rate of 10 °C/min), during which melting curves were recorded. All tests were carried out under a dry nitrogen atmosphere at a pressure of 0.05 MPa, and the flow rates of purge gas and protective gas were 80 mL/min and 30 mL/min, respectively. The chemical structures of GO were characterized by Fourier transform infrared spectroscopy (FT-IR, Thermo Scientific Nicolet iS20, Waltham, MA, USA). Their morphologies were observed with a scanning electron microscope (SEM, Zeiss Sigma 300, Munich, Germany). The surface characteristics were analyzed by atomic force microscope (AFM, Bruker Dimension Icon, Saarbrücken, Germany).

#### 2.3.2. Characterization of the Prepared Membranes

The morphology of the cross section and surface for various membranes was observed with a scanning electron microscope (SEM Gemini 300, ZEISS, Jena, Germany) with an accelerating voltage of 15 kV. (The upper surfaces and cross-sectional morphologies of various membranes were examined using scanning electron microscopy (SEM Gemini 300, ZEISS, Jena, Germany) with an accelerating voltage of 15 kV.) When characterizing the cross-sectional morphologies, the samples were prepared by fracturing membranes after fully cooling in liquid nitrogen. All the samples were treated with Au/Pd sputtering and carefully handled to avoid contamination.

X-ray photoelectron spectroscopy (XPS Thermo Scientific K-Alpha, Waltham, MA, USA) was used to investigate the surface chemical compositions of membranes. The range of survey spectra is from 0 to 1300 eV and the C1s peak of high-resolution spectra were detected. XPS full-scan spectra were recorded within the range from 0 to 1300 eV with a pass energy of 150 eV with a monochromatic Al Kα X-ray source at 1486.6 eV.

Fourier transform infrared spectroscopy-attenuated total reflectance (FTIR-ATR) measurement was carried out using a Thermo Scientific Nicolet iS20 Fourier transform infrared spectrometer. The samples were placed on the sample holder and all spectra were recorded from 4000 cm^−1^ to 600 cm^−1^.

The roughness of each membrane surface was examined by atomic force microscopy (AFM) (Bruker Dimension Icon, Saarbrücken, Germany) at room temperature.

The crystal structure of polyamide 66 in the prepared membranes was determined by means of a wide-angle X-ray diffractometer (WAXD, DX-2700, Shanghai, China). The scanning parameters included the source intensity (40 kV/30 mA), λ (1.54 Å, Cu Kα line), source slit width (0.6 mm), increment rate (1.20°/min), and scanning range (5–80°).

The melting and crystalline properties of PA66/GO membranes were studied using a differential scanning calorimeter (DSC 200-F3, NETZSCH, Selb, Germany) in a nitrogen atmosphere. About 8 mg of dry membrane was sealed in an aluminum pan. The test was performed following the same cool–heat procedure as described in Section 2.3.1. The degree of crystallinity was calculated based on the following equation:(1)$$Xc=\frac{\Delta Hf}{\Delta Hf^{*}}$$
where *Xc* is the degree of crystallinity (%); ΔH*_f_* and ΔH*_f_** represent the fusion enthalpy (melting enthalpy) of the membrane and PA66 with 100% crystallinity, respectively; The value of ΔH*_f_** is 188 J g^–1^.

#### 2.3.3. The Hydrophilicity of Membrane

The hydrophilicity of the membrane was observed based on the water contact angle measurement (WCA). The WCA of the membrane was measured using an optical contact angle tester (OCA15Pro, DataPhysics Instruments GmbH, Filderstadt, Germany) according to the sessile-drop method. This contact angle can be described as an angle between the sample surface and calculated drop shape function, the projection at which the drop image is referred to as the baseline. Briefly, a water droplet (about 10 μL droplet) was deposited on the membrane surface and the value of contact angle was observed and recorded until there was no change of droplet during the short measurement period. The average was made by measuring parallel three times for each sample.

#### 2.3.4. The Permeability, Anti-Fouling Performance for the Membrane

Pure water flux (PWF) was measured with a cup ultrafilter at 0.15 MPa pressure. The flux was equilibrated for the passage of the first 30 min permeate whilst the following 10 min permeate was collected. Pure water flux was evaluated by: (2)$$J=\frac{V}{A\cdot \Delta t}$$
where *J* is pure water flux (L/(m^2^·h)), *V* the volume of penetrated water (L), *A* the effective area of the membrane (m^2^) and Δ*t* is the recorded time (h). All the experiments were conducted thrice to obtain the results presented in this study, which were an average value.

UV/visible spectroscopy (TU-1901, General analysis) was employed to measure the concentration of the feed solution and permeation of the BSA solution at the wave-length of 280 nm. The feed solution was 0.1 g/L BSA solution. Then, the filtered solution was obtained in the same way as water flux testing. The membrane rejection (*R*) was obtained by Equation.
(3)$$R=\frac{C_{F}-C_{P}}{C_{F}}\times 100\%$$
where *R* is the percentage retention rate or protein (BSA) rejection, *C_F_* is the concentration of feed solution and *C_p_* is concentration of permeation.

The anti-fouling property of the PA66/GO blend membranes compared to pure PA66 membrane was investigated using BSA. This protein was selected as the model foulant because it is one of the common foulants of membranes. The foulant (Feed) solution was also 0.1 g/L BSA solution. The protocol for the fouling tests involved four main stages, which included the recording of the water flux at 0.15 MPa, subjecting the membrane to a feed solution for one hour, backwashing with pure water for one hour, and calculating the flux recovery rate (*FRR*) by Equations.
(4)$$FRR=\frac{J_{W2}}{J_{W1}}$$
where *J*_W2_ is the PWF after washing of the membrane, *J*_W1_ is the initial PWF before fouling the membrane with BSA.

#### 2.3.5. The Porosity of the Membrane

After measuring the dry weight of the membranes, they were immersed in n-butanol for 12 h to become wet, and the wet weight was then measured. The porosity for the membranes was obtained by Equation.
(5)$$\varepsilon =1-\frac{W_{1}/\rho_{m}}{(W_{0}-W_{1})/\rho_{d}+W_{1}/\rho_{m}}\times 100\%$$
where *ε* is the porosity (%), *W*_1_ is the mass for the dry membrane (g) and *W*_0_ is the mass of membrane (g) after absorbing n-butanol. *ρ_d_* and *ρ_m_* represented the density of n-butanol (g cm^−3^) and membrane (g cm^−3^), respectively.

#### 2.3.6. Mechanical Property of Membrane

The microcomputer controlled electronic universal testing machine (CMT8501, China) was employed to determine the breaking strength and breaking elongation for the membrane by measuring the stress–strain curve. The size of the selected membrane was 70 mm × 10 mm and the distance between the two clamps was 30 mm. Each sample was tested three times to obtain average strength. They were obtained by Equations (6) and (7), respectively.
(6)$$R=\frac{F}{S}$$
where *R*, *F* and *S* are the breaking strength (MPa), the breaking force (N) and the area of cross section for the membrane (mm^2^), respectively.
(7)$$\varepsilon =\frac{L}{L0}$$
where ε, *L* and *L*_0_ represented the break elongation (%), final length (mm) and initial length (mm), respectively.

## 3. Results and Discussion

### 3.1. Characterization of GO

The GO was characterized in several ways. First, the time nano industrial graphene oxide powder (GO) was tested by XRD to verify its purity, as shown in Figure 2. It showed a significant diffraction peak at 2θ of approximately 10° that corresponded to GO and is consistent with the results in the literature. This confirmed the high purity of GO used in this study [11].

Secondly, to characterize the thermal stability of the GO material, DSC measurements were performed. Figure 3 shows the DSC curves from the thermogravimetric analysis of GO. A clear peak of heat absorption was observed on the DSC curve at around 190 °C, which is consistent with that reported in the literature. However, the weight loss peak did not appear again at around 300 °C, which may be caused by the rapid evaporation of a large amount of water present between the GO sheets.

Thirdly, Figure 4 shows the FTIR spectrum of GO nanoparticles. The absorption peaks on the spectrum are –OH stretching (3417.09 cm^−1^), C=O stretching in the carboxylic acid (1730.31 cm^−1^), C=C stretching (1585.62 cm^−1^), C–OH stretching (1188.11 cm^−1^), and C–O stretching (1049.68 cm^−1^), respectively, which confirms that GO contains a large number of oxygen-containing functional groups, and the presence of hydroxyl groups can form hydrogen bonds to enhance the interaction between the nanosheet and the membrane surface [12]. As can be seen from Figure 4b, there are a large number of wrinkles on the surface of the GO sheet, which are caused by the destruction of the crystal structure of GO, due to the large number of functional groups existing between the sheets. From the AFM figure, it can be clearly seen that the GO monolayer is sheet and slightly rotated at the edge.

The GO/PA66 blend membranes were prepared by the immersion phase inversion process, using water as a coagulant. When using this method, the solvent and non-solvent had good miscibility that would be used for the phase inversion of the PA66 solutions via the solvent–non-solvent exchange process [14]. In this study, FA and PC were chosen as solvents due to their strong interactions with polymers and miscibility with water. Moreover, propylene carbonate, as an organic solvent with excellent performance, was a good promoter of many insoluble substances. One of the common non-solvents, so called coagulants, was water. Beyond water, sometimes alcohols, such as ethanol and propanol, can also be used as coagulants.

In order to fabricate the GO/PA66 blend membranes, it was necessary to make a homogeneous GO solution in FA and PC before blending them with the PA66 solution in FA and PC. To further explore the dispersion of GO in various polar organic solvents, GO suspensions were prepared. Figure 4 shows the absorption spectra of UV/visible spectrophotometry of GO solutions in various polar organic solvents. It is well known that the more stable suspension of GO in solvents shows increased absorption in UV/visible spectrophotometry [15,16]. Based on the explanation in the previous paper, Figure 5, it was found that PC was the best solvent for the dispersion of GO, and the second was FA, the third FA and PC, the last one being DI water [14]. The low dispersion of GO in acidic solutions can be attributed to the low pH value of the solution, which reduces the charge at the edges of GO, thus increasing the insolubility of the flakes, and the monolayer structure becomes inhomogeneous, as shown in Figure 6. The results showed that the GO used in this study had a certain degree of hydrophobicity. This may be due to the hydrophobicity of the central main structure in the graphene sheet and the weak hydrophilicity of the edge of the sheet. With the increase of the GO doping amount, the hydroxyl groups on its surface were further esterified and weak hydrophilicity was further enhanced. So, in this study, PC was found to be the best solvent for the dispersion of GO.

### 3.2. Chemical Characteristics of the Membranes

To explore chemical compositions of membrane surfaces, XPS was performed [1].

The XPS full-scan spectrum (Figure 7a) showed both the pure PA66 membrane and the PA66-GO membrane contain oxygen (531 eV), nitrogen (399 eV), carbon (285 eV), and their atomic contents are listed in Table 2 respectively. Oxygen, nitrogen and carbon were inherent elements of polyamide 66 and GO [12]. From the spectral graph of C1s, for pure P66 membrane (Figure 7b), the spectra can be deconvoluted into three peaks located at 284.8 eV, 285.9 eV and 287.8 eV attributing to C–C, C–N and N–C=O groups [17]. After the incorporation of GO into PA66 membrane, the same result can also be obtained (Figure 7c). It was found that the peak strength of the C=O group in Figure 7e was much higher than that in Figure 7d, indicating that GO was fixed in PA66 membrane and that the mutual cross-linking degree between the two was further improved, and the adhesion between GO and PA66 was further enhanced; which was conducive to improving the stability of the membrane. Moreover, the content of the C=O bond increased by 4.6%, which further proved the effective combination of the two [18,19]. This was not only due to the hydrogen bond connection between groups, but also possibly because the amide group changed the ring-opening degree of GO to form some kind of effective covalent bond, which could effectively reduce the interface resistance generated between the two groups.

Furthermore, FTIR was used to compare and analyze the changes of the surface chemical composition of GO-doped membranes before and after the reaction [1]. It can be clearly seen from Figure 8 that the ultrafiltration membranes before and after the GO reaction contain a large number of related functional groups. Both spectra showed the existence of N–H stretching vibration (3295.52 cm^−^^1^), N-H shear vibration and C–N stretching vibration (1533.73 cm^−^^1^), C=O stretching vibration (1629.42 cm^−^^1^), and C–N stretching vibration (1416.01 cm^−^^1^). The characteristic peaks at 2932.9 cm^−^^1^ and 2858.27 cm^−^^1^ all correspond to the stretching vibration of methylene (–CH_2_). In addition, small peaks at 1310–1200 cm^−^^1^ correspond to C–N stretching vibration and N–H shear vibration in polyamide. The peak strength at 1629.42 cm^−^^1^ was further enhanced compared to the pure membrane, which was attributed not only to the C=O stretching vibration in the polymer, but also to the C=C stretching vibration in the benzene-like structure of GO, which indicated that GO was successfully dissolved in the polyamide system [20,21,22]. In fact, the C=C peak had shifted from 1630 cm^−^^1^ (in the pure GO) to 1629.42 cm^−^^1^ (in PA66/GO), due to formed hydrogen bonds (C=O· · ·H–O) [23]. These results were consistent with those obtained by XPS and affected the performance of the membrane.

### 3.3. Morphology of the Membranes

The upper surface of PA66 and PA66/GO membranes are displayed in Figure 9. The axial growth of the petal lamellae was identified in the magnified view near a crevice (Figure 9a), and the layers crossed in series with each other. As described in the literature, the upper surface of the membrane could be considered to be composed of interlocking bundles of aggregates, showing the characteristics of terminal splaying. This may be caused by the fact that during the crystallization process, a large number of nucleated embryos were immersed in the water bath. Due to the short phase separation time, the solvent formic acid was not displaced outward in time and diffused inward [10,24]. The shape of the axial crystals indicates that they are in the early stage of mature spherulites, which is typical of the sheaf-like appearance of metaphase spherulites.

However, for PA66/GO membranes, as shown in Figure 9b–d, the surface mutual cross-linkage further increased, the macropores are reduced, the end flakes are much smaller, and the end sectors are further spheroidized, showing a porous spherical shape. This is because the doping of GO increases the viscosity of the casting solution on the one hand, and increases hydrogen bonding formed by oxygen-containing groups on the other hand, leading to the weakening of the interaction between the polymer and diluent during the aging process of the casting solution and the reduction of the voids between the grains [19,25]. With the increase of the GO doping amount, the polymer concentration increased and the membrane surface became more flocculated.

The cross-sectional image of the membrane is shown in Figure 10. There were obvious differences in morphology and structure between the PA66 membrane and the PA66/GO membranes. As can be seen in Figure 10, the pure PA66 membrane was composed of large and complete axial crystals and tended to open outwards. These crystals were much larger than those on the surface of the membrane, and the crystal column was more robust, which was related to the strong stability of PA66 itself and maintained the supporting stability of the membrane. When GO was doped in PA66, the PA66/GO membranes became densified, the gaps between crystals were filled and compressed, and the fan between crystals was more closely connected [10]. As the GO concentration increases, the membrane becomes denser. The oxygen-containing functional groups on GO form hydrogen bonds with amide groups, which enables better dispersion of GO in polyamide systems. The cross-sectional images also confirmed this result.

Figure 11 shows the 3D AFM images of the upper surface of the substrate upon the addition of GO. As can be seen from the figure, the PA66/GO membranes exhibited a significantly smoother surface compared with the PA66 substrates [11]. This may be due to the fact that the polar groups on GO effectively increase compatibility with the polymer, allowing it to further act as a reinforcing filler material to be stably dispersed in the polymer material, reducing the degree of undulation and fluctuation on the membrane surface. This kind of reinforcing filling characteristic avoids the defects of large pores in the membrane, improves the interception ability of the membrane, and is beneficial to improving the filtration performance of the membrane. The high surface roughness of the PA66 membrane may be due to the ridge and valley structure formed between the two monomers of adipic acid and hexamethylamine in the PA66 material itself.

In addition, the Ra value on the membrane surface increased from 60.7 nm to 90.7 nm with the increase of GO loading from 0.1 to 0.5 wt.% in the doped solution. This increase in roughness may be due to the increase in the degree of agglomeration of GO nanomaterials on the membrane surface [19,26]. Although the roughness value increases, the PA66/GO membranes still remain smooth, which is conducive to anti-fouling.

### 3.4. Crystalline Properties of the Membranes

In order to further obtain information related to the polymorphism of PA66 membranes in this study, XRD measurements were conducted [27] and Figure 12a shows the diffraction patterns of PA66 membranes prepared with different GO contents. There are two diffraction peaks at 2θ values of about 20.1 and 23.9, which correspond to crystal types α1 (100) and α2 (010/110) of PA66, respectively, proving that crystal type α is formed during the membrane forming process. Moreover, the distance between the two diffraction peaks slowly increased with increase of GO additives, indicating that the crystalline spacing of PA66/GO membrane slightly increased and the crystal structure of the polymer became imperfect. The hydrogen bond originally formed between polyamide molecules is weakened by the addition of GO oxygen-containing functional groups, thus increasing the flexibility of the molecular chain.

The DSC curves of PA66 membranes prepared with different GO addition amounts are shown in Figure 12b. As can be seen from the P0 curve, the glass transition temperature of pure PA66 membrane is around 49 °C, while the crystallization temperature of PA66/GO membranes decreases with increase of GO loading when GO is added, indicating that GO improves the crystallization behavior of PA66 to some extent and can increase its crystallization rate [18,27].

### 3.5. The Contact Angle of the Membranes

In order to explore the effect of GO doping on the hydrophilicity of the membranes, the contact angles of various membranes in this study are shown in Figure 13. It is reported that the contact angle of the membrane surface decreases significantly after GO doping, which is due to the hydrophilic nature of the embedded GO nanomaterials. The pure PA66 membrane has the largest contact angle because the planar sawtooth shape of the PA66 molecule has a certain rough structure, and the resulting membrane surface has high roughness, but it is still a hydrophilic membrane. However, the contact angle increases as GO changes from 0.1 to 0.5 wt.%, which is due to the increased agglomeration of more GO on the effective surface of the membrane, increasing the contact area of water with the membrane surface [12].

### 3.6. The Permeability of the Membranes

When the incorporation of GO increases from 0 to 0.5 wt.%, the porosity of the ultrafiltration membrane decreases from 53.2 to 23.8% and presents an overall downward trend, while P1 and P2 membranes have very similar porosity values, as shown in Figure 14. It can be explained that, with the addition of GO, the viscosity of the mixed casting solution would gradually increase, thereby hindering the phase separation process and resulting in a final membrane with a relatively dense structure.

Figure 14 also shows the effect of GO on PWF and BSA rejection of the PA66 ultrafiltration membrane. It is worth noting that compared with a pure PA66 ultrafiltration membrane, the permeability of the doped GO ultrafiltration membrane to pure water is lower, decreasing from the initial 16.3 to 3.5 L/m^2^·h. The further reduction of water flux with high GO doping can be explained by the significant change of the contact angle on the membrane surface and the densification of the membrane as a result of higher resistance to the permeation of water molecules [28]. Finally, the decrease of water flux of the ultrafiltration membrane may be due to the combined effect of the strong hydrophobicity of the central structure of GO and the weak hydrophilicity of the oxygen-containing groups at the edge.

Conversely, the BSA rejection rises with the increase of GO content, reaching a maximum of 92.5% when the GO amount was 0.5 wt.%. This is because GO migrates not only to the surface of the membrane, but also to the pore wall of the membrane, and is evenly dispersed to optimize the pore size, which is consistent with the morphology shown in Figure 9 and Figure 10. At a certain pressure, the decrease of membrane pore size increases the resistance of BSA solution to passing through the membrane pore. It can be said that GO changes the structure of the membrane to a certain extent. In this way, the porosity of PA66 membrane has a high rejection rate, which further enhances the ability of the membrane to select contaminants.

### 3.7. Anti-Fouling Performance of Developed Membranes

The anti-fouling performance of various PA66 ultrafiltration membranes was studied, and the flux recovery rate was taken as an important parameter to examine the water flux recovery of the membranes after protein permeation. According to Figure 15, the flux recovery rate (*FRR*) of the PA66 membrane without GO is only 67.3%. The reason is that the larger pores of the pure PA66 membrane can easily accumulate foulants without the strong shear force between water and accumulating proteins when water permeates. Interestingly, after GO was added into the membrane, the *FRR* value increased significantly, and it continued to rise with the increase of GO, up to 91.3%, which was consistent with the results of the filtration performance. One of the key reasons is that the strong electrostatic repulsion generated by GO surface chemistry creates an energy barrier for the adsorption of BSA onto the membrane surface. On the other hand, the oxygen-containing groups on GO can interact with water molecules through Van der Waals forces and hydrogen bonds, forming exclusive water molecular channels in membrane pores to avoid protein deposition and adsorption [29,30]. The membrane screening mechanism is graphically shown in Figure 16. Therefore, GO plays an important role in improving the antifouling performance of PA66 ultrafiltration membrane.

Table 3 summarizes a series of ultrafiltration membranes used to improve membrane anti-fouling performance. It can be found that the flux recovery rate of almost all membranes is above 90%, indicating that these membranes have excellent performance in anti-fouling, which can provide a reference direction for the practical application of membranes with high anti-fouling performance. The ultrafiltration membranes prepared in this study also have good anti-pollution properties.

### 3.8. Mechanical Performance of the Membranes

Consistent with our expectations, the mechanical properties of the membrane gradually increased with an increase of GO. As shown in Figure 17, before and after GO loading, the tensile strength and elongation at break of the membrane ranged from 12.4 MPa to 24.3 MPa and 101.3% to 106.5%, respectively. This is because the addition of GO can improve the mechanical strength of the membrane and make the structure of the membrane more compact [33].

## 4. Conclusions

The effect of GO nanomaterial on polyamide 66 ultrafiltration membranes was studied in this paper. A series of PA66 ultrafiltration membranes, with and without GO, were prepared by the immersion precipitation phase conversion method and conclusions made as follows:(1)When GO is added to PA66 ultrafiltration membrane, the physicochemical properties of the membrane can be improved from the aspects of microporous structure, hydrophilic properties, surface roughness, and anti-fouling performance.(2)SEM results show that after the addition of GO, the PA66 membrane becomes denser, and the crystals change from large and sharp axial crystals to small and round spherical crystals in the process of membrane formation. Pure PA66 membrane has larger and more pores than PA66/GO membrane.(3)After loading GO into PA66, the surface of the membrane becomes smooth and the surface morphology changes significantly. However, with further increase of GO, the roughness of the ultrafiltration membrane increases from 60.7 nm to 90.7 nm, which is due to the agglomeration of GO.(4)The contact angles of all PA66/GO membranes are smaller than the pure PA66 membrane, because of the hydrophilic nature of the oxygen-containing groups on GO. However, when the amount of GO increases, the contact angles increase because the central structure of GO is hydrophobic.(5)The anti-fouling performance of the membrane was significantly improved after incorporating GO into membranes, which benefited from the optimization effect of GO on the membrane structure, and the flux recovery rate was up to 91.3%.

## Figures and Tables

**Figure 1 membranes-12-00458-f001:**
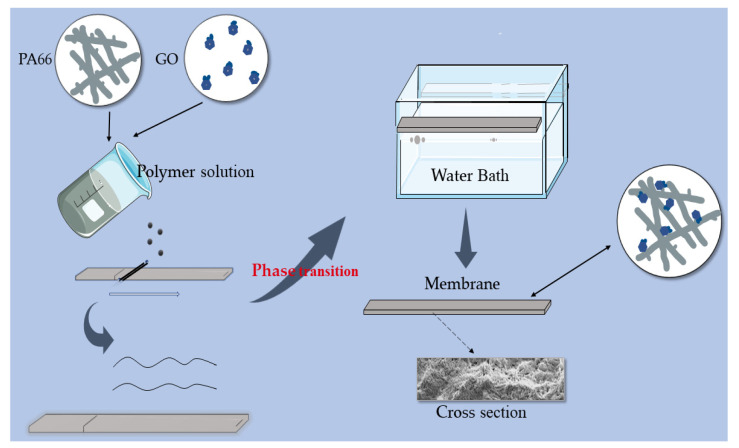
Schematic diagram of membrane formation mechanism.

**Figure 2 membranes-12-00458-f002:**
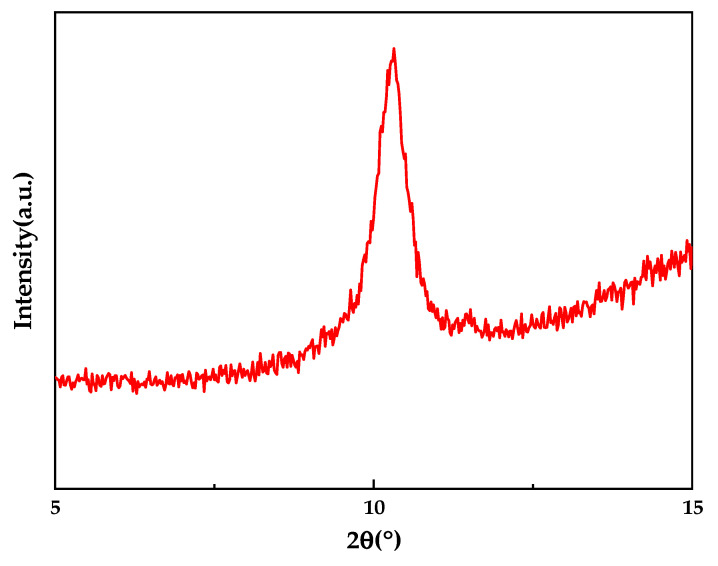
XRD spectrum of GO.

**Figure 3 membranes-12-00458-f003:**
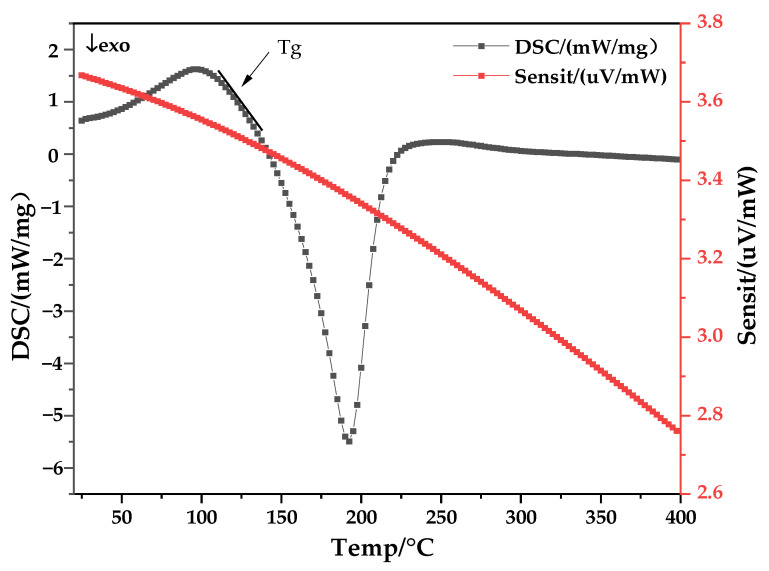
Thermogravimetric scanning calorimetry and differential thermal curves of GO.

**Figure 4 membranes-12-00458-f004:**
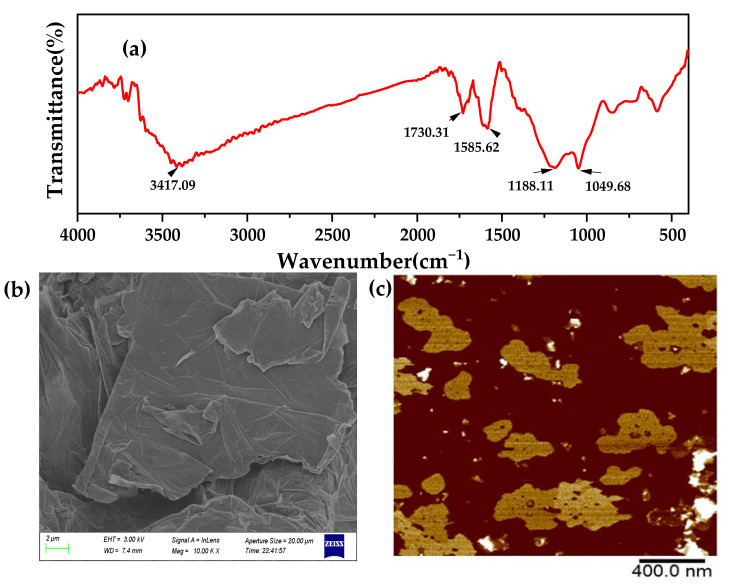
FTIR (**a**) spectra of the GO nanoparticles; SEM image of the GO (**b**); AFM image of the GO (**c**).

**Figure 5 membranes-12-00458-f005:**
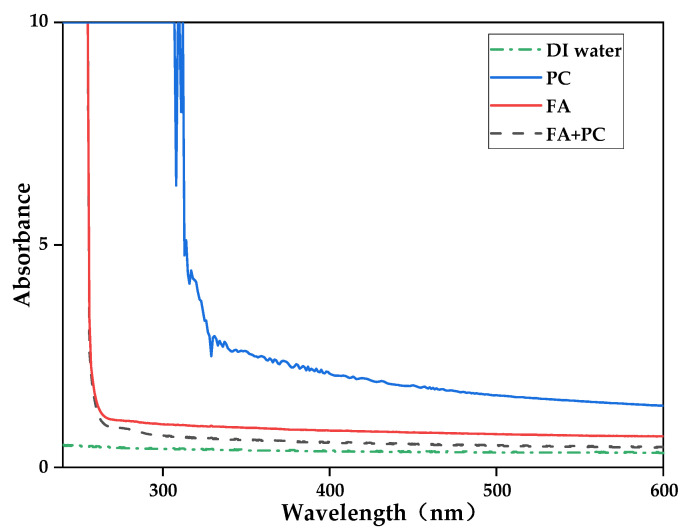
UV–vis spectra of GO dispersed in different solvents.

**Figure 6 membranes-12-00458-f006:**
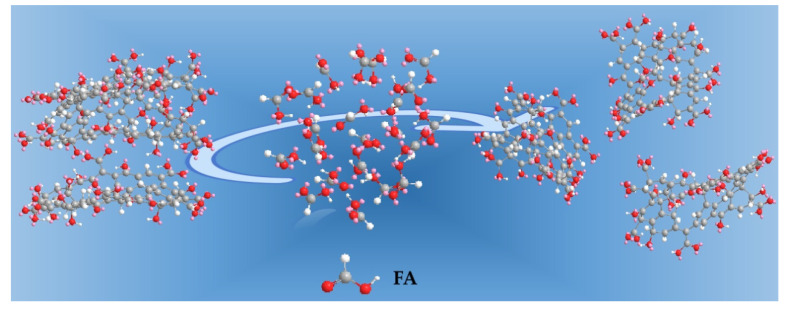
Distribution of GO in acidic solvents.

**Figure 7 membranes-12-00458-f007:**
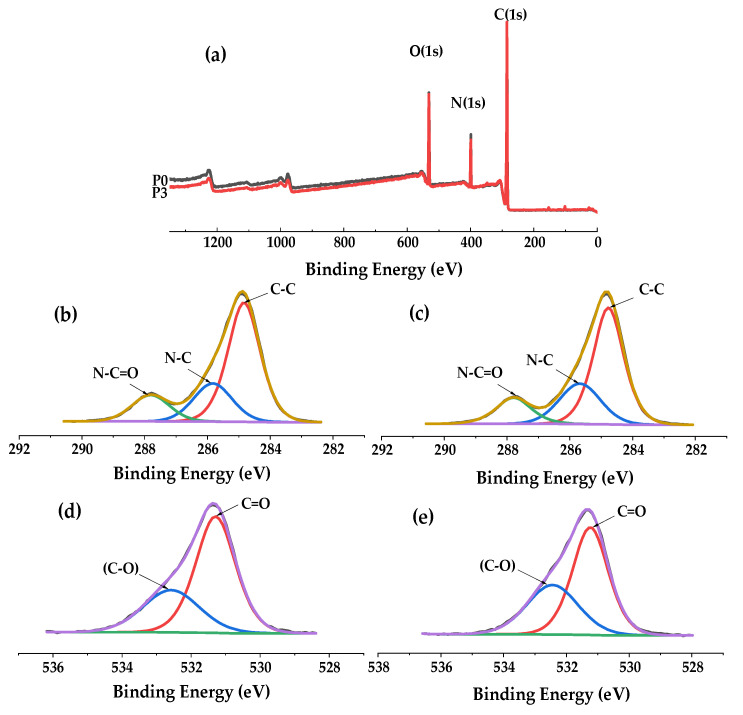
(**a**) XPS full-scan spectra of P0 and P3 membrane; (**b**) and (**c**) C1s spectra of P0 membrane and P3 membrane; (**d**) and (**e**) O1s spectra of P0 and P3 membrane.

**Figure 8 membranes-12-00458-f008:**
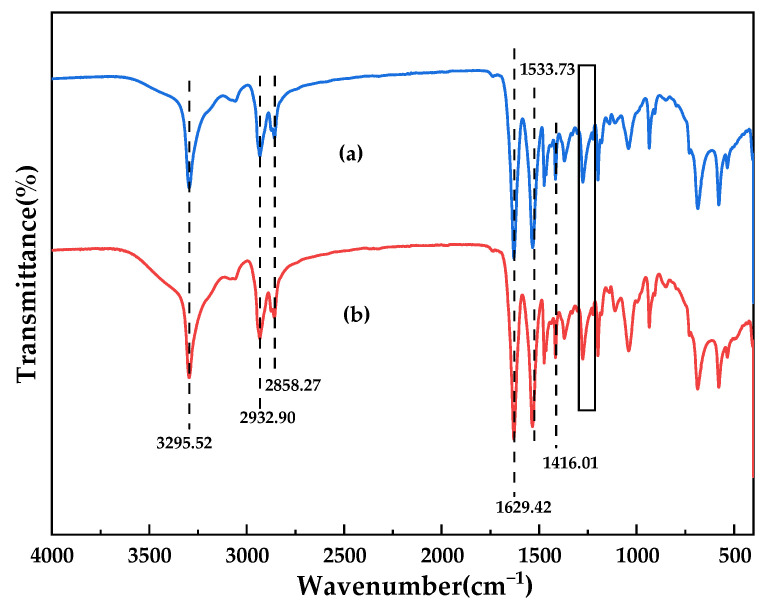
FTIR spectra of PA66 membrane before and after modification by GO. ((**a**) P0 membrane; (**b**) P3 membrane).

**Figure 9 membranes-12-00458-f009:**
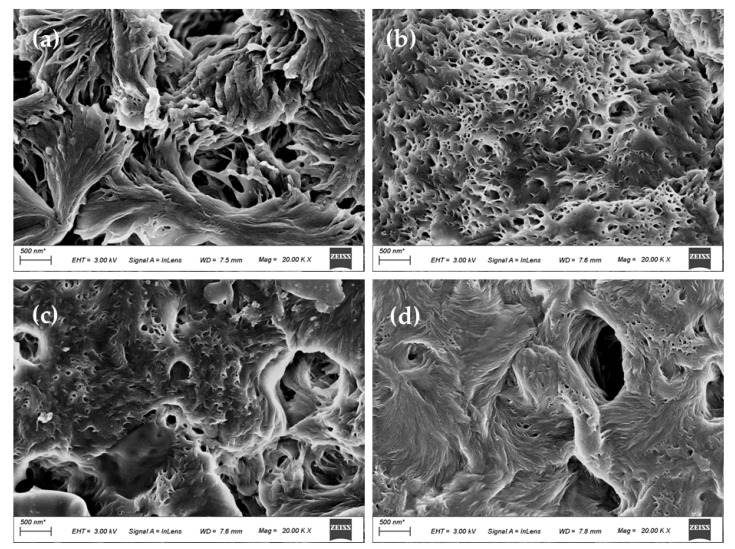
SEM images of upper surfaces of PA66 and PA66/GO membranes, (**a**) P0, (**b**) P1, (**c**) P2 and (**d**) P3.

**Figure 10 membranes-12-00458-f010:**
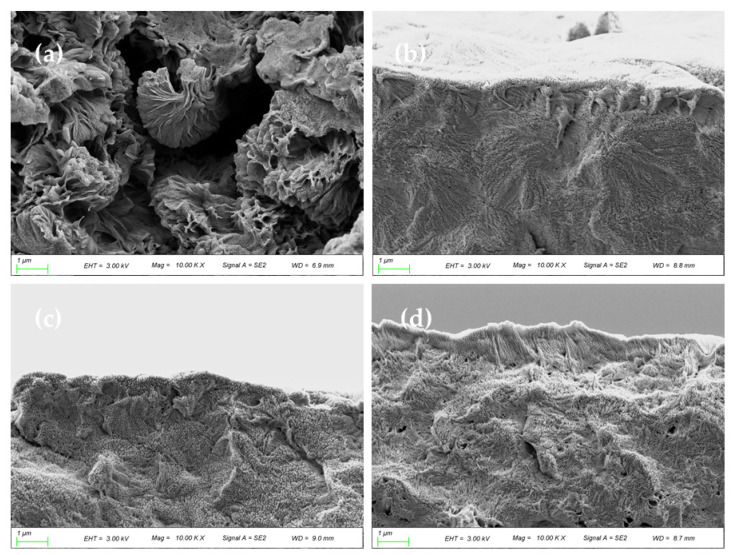
SEM images of cross-section of PA66 and PA66/GO membranes, (**a**) P0, (**b**) P1, (**c**) P2 and (**d**) P3.

**Figure 11 membranes-12-00458-f011:**
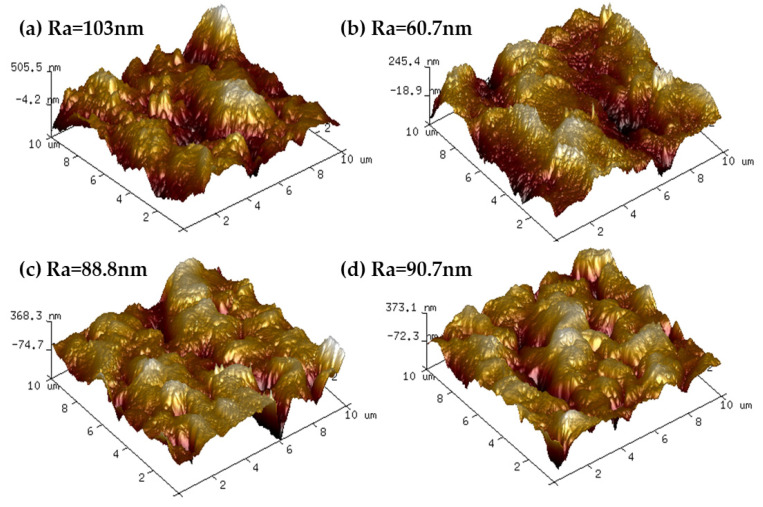
AFM images of PA66 and PA66/GO membranes, (**a**) P0, (**b**) P1, (**c**) P2 and (**d**) P3.

**Figure 12 membranes-12-00458-f012:**
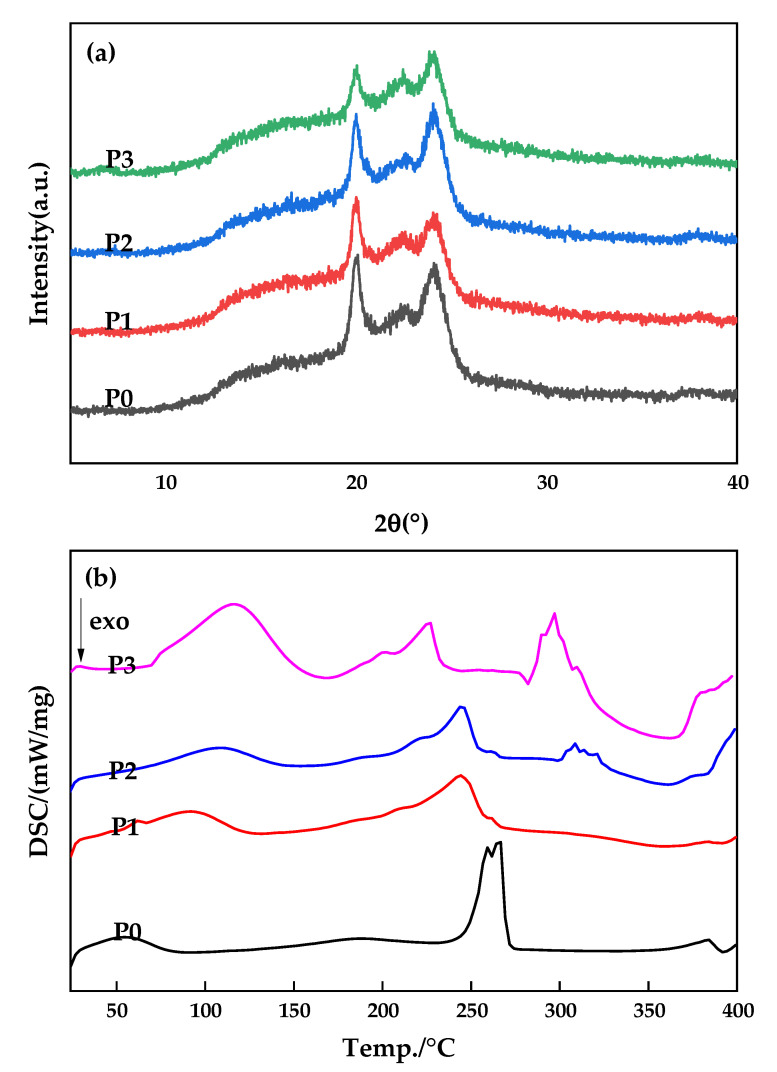
X-ray diffraction patterns (**a**) and DSC curves (**b**) of membranes obtained with different GO additives.

**Figure 13 membranes-12-00458-f013:**
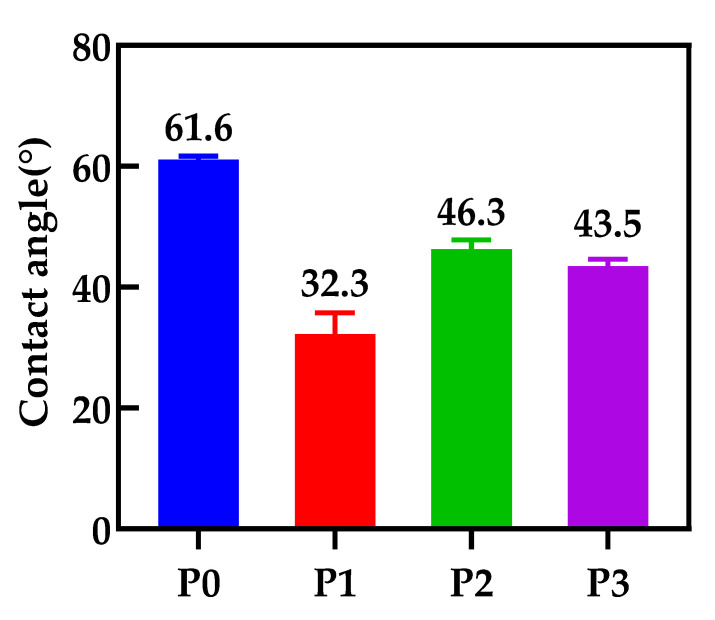
Contact Angle of membranes with different GO doping.

**Figure 14 membranes-12-00458-f014:**
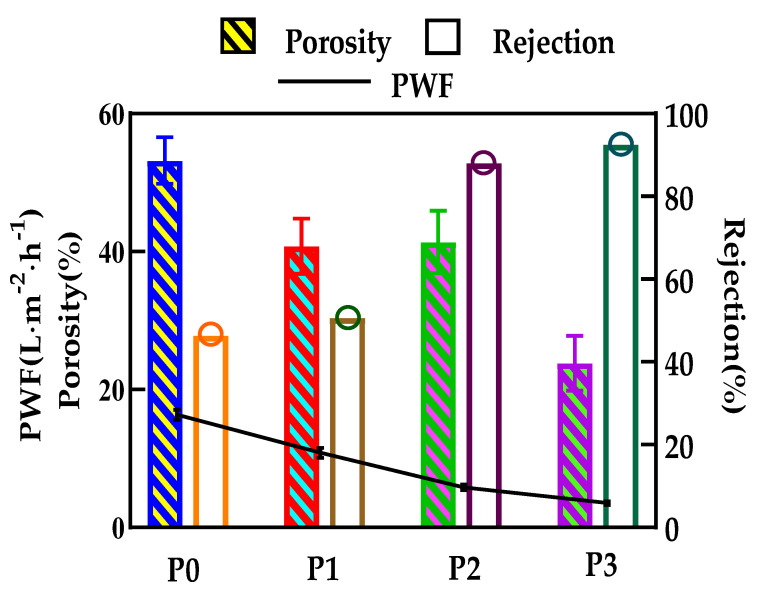
Pure water flux, porosity and rejection of ultrafiltration membranes with different GO doping amounts.

**Figure 15 membranes-12-00458-f015:**
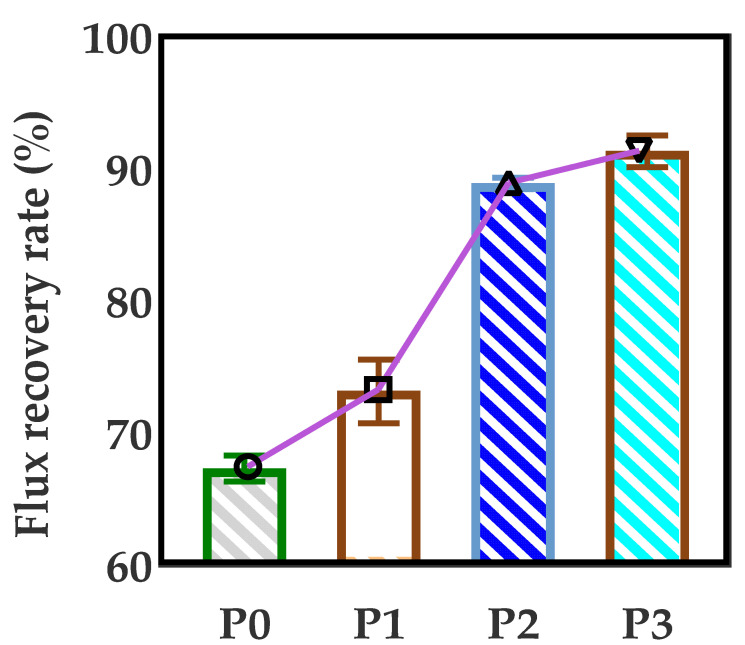
Flux recovery rate of PA66/GO membrane varies with GO content.

**Figure 16 membranes-12-00458-f016:**
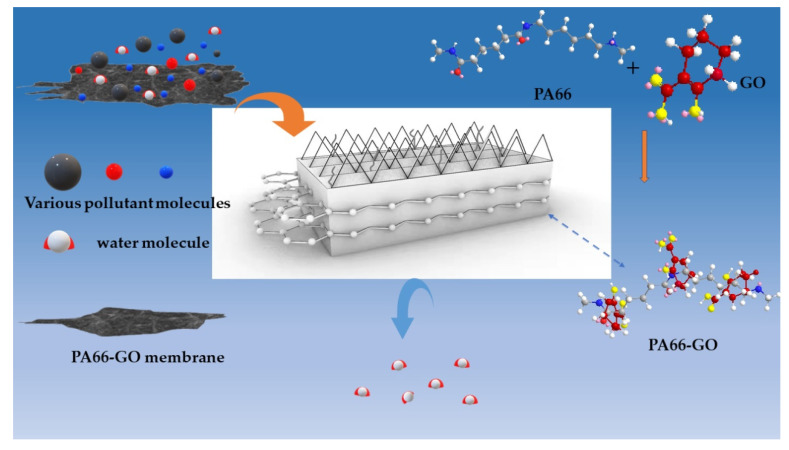
Schematic diagram of anti-fouling mechanism.

**Figure 17 membranes-12-00458-f017:**
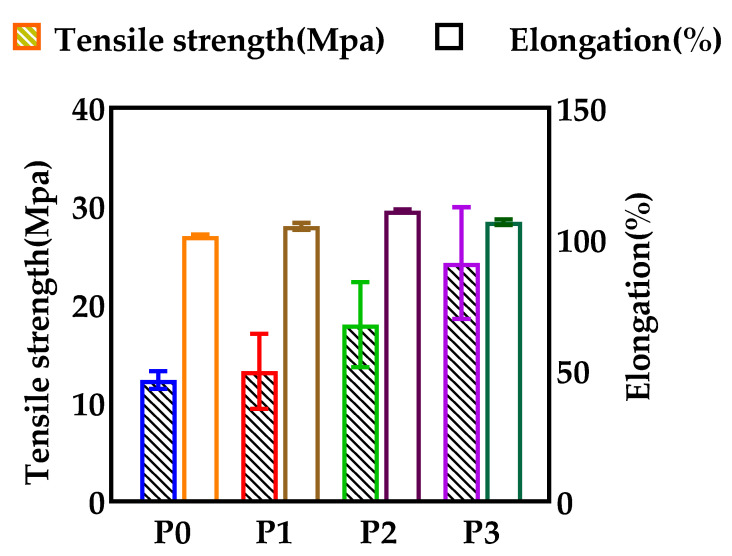
Tensile strength and elongation at break of different GO doped membranes.

**Table 1 membranes-12-00458-t001:** Casting solution composition and coding of different membranes.

Code	Membrane	Content (wt.%)			
		PA66	GO	FA	PC
P0	PA66	22	0	68	10
P1	PA66-GO	22	0.1	68	9.9
P2	PA66-GO	22	0.3	68	9.7
P3	PA66-GO	22	0.5	68	9.5

**Table 2 membranes-12-00458-t002:** Elemental composition by atomic percent of pure PA66 membrane and PA66-GO membrane.

Membrane	C (%)	N (%)	O (%)
PA66 (P0)	73.7	11.6	14.7
PA66-GO (P3)	74.2	10.6	15.2

**Table 3 membranes-12-00458-t003:** Comparison of the reported UF membranes with the UF membrane prepared in this study.

Membrane	Conditions	Anti-Fouling Performance (*FRR*)	Flux (L/(m^2^·h))	Reference
	0.1 MPa			
GO+PVP+PVDF	48.32 cm^2^(area)	90.5%	104.3	[1]
	1 g/L BSA			
	0.1 MPa			
DA+DMAPAPS+PVDF	33.18 cm^2^	96.3%	364	[9]
	1 g/L BSA			
	0.2 MPa			
Cu_2_S+PVDF	-	92.4%	248.25	[5]
	0.5 g/L BSA			
	0.1 MPa			
Ar/O_2_+W_50_Ni_25_B_25_+PSf	-	91.3%	321.5	[8]
	0.6–0.8 l/m BSA			
	0.1 MPa			
GO-Fe_3_O_4_+FAS+PSf	13.4 cm^2^	98.2%	323.2	[31]
	1 g/L BSA			
	0.15 MPa			
SMA+PSf	-	91%	147	[26]
	1 mg/mL BSA			
	0.4 MPa			
AM-MA+PA6	-	91.1%	19	[32]
	- BSA			
	0.1 MPa			
GO+PVDF	-	85.1%	163	[22]
	1 g/L BSA			
	0.1 MPa			
Pluronic F127+PES	28.7 cm^2^	94.12%	140	[30]
	1 g/L BSA			
	0.15MPa			
GO+PA66	37.4 cm^2^	91.32%	3.5	This work
	0.1 g/L BSA			

## Data Availability

The data used to support the findings of this study are available from the corresponding author upon request.

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
