# Peer review of "Graphene Oxide Modified Polyamide 66 Ultrafiltration Membranes with Enhanced Anti-Fouling Performance"

_membranes, 2022, doi:10.3390/membranes12050458_

Round 1
Reviewer 1 Report
Author did not mentioned the amount of water(mL) drops in Contact angle measurement.
When added the GO in PA66 then contact angle decreases, support the good literature.
below 90 degree contact angle membrane have hydrophilic nature
Is hydrophilic film, suitable for membrane, antifouling property
Check the whole manuscript for contradiction in membrane property.
Reviewer 2 Report
Presented article shows the results of preparation, characterization and application of a novel asymmetric polyamide 66 composite membranes with GO as a filler in ultrafiltration process. The topic of the paper is interesting and the presented graphs, figures and Tables are very clear. I recommend to publish this work after minor revisions and complements.
Introduction
Please add information about other composite membranes used in the ultrafiltration process. What fillers besides GO have been used in ultrafiltration membranes, so far? What are the advantages of using such fillers?
Figure 10
Please increase the font size of the axis description
Figures
The font size on each graph is different. Please standardize the font size.
Comparison with literature data
Please, compare investigated membranes with other composite membranes, reported in literature, used in the ultrafiltration process.
Reviewer 3 Report
The manuscript, entitled “Graphene oxide modified polyamide 66 ultrafiltration membranes with enhanced anti-fouling performance” is an interesting paper. According to me, this manuscript merits to be accepted in the journal Membranes. However, the below changes should be addressed before taking it further.
Abstract:
- Please, improve the abstract in the manuscript (More results must be added to attract the readers).
Introduction:
- The introduction should be improved.
- Please, discuss the different types of polymers used in membrane separation then, discuss your reasons for choosing the Polyamide 66.
- Line 46; AIE (please, write the full name of abbreviation).
- Please, discuss the different types of nanomaterials and their advantages then, discuss your reasons for choosing GO nanosheets.
- Please, improve the aim of the study of this paper.
Results and discussion
- The standard deviation should be mentioned in all of the paper.
- FT-IR, SEM, and AFM of GO should be added.
- The salt rejection of membranes could be added
- The water content of the membranes could be studied.
- Antifouling study; antimicrobial resistance could be studied.
- Antifouling study; additional proteins and amino acids could be studied.
- The authors should relate their findings to other research results.
Round 2
Reviewer 3 Report
The paper can be accepted in the present form